# The Phenotypical Profile and Outcomes of Neonates with Congenital Tracheoesophageal Fistula Associated with Congenital Cardiac Anomalies Presenting for Surgery

**DOI:** 10.3390/children9060887

**Published:** 2022-06-14

**Authors:** Nomvuyo Hoyi, Palesa Mogane, Nthatheni Madima, Palesa Motshabi

**Affiliations:** Department of Anaesthesiology, Faculty of Health Sciences, School of Clinical Medicine, University of the Witwatersrand, Johannesburg 2193, South Africa; moganep@gmail.com (P.M.); nthatheni@icloud.com (N.M.); palesa.motshabi@wits.ac.za (P.M.)

**Keywords:** tracheoesophageal fistula, congenital cardiac anomalies, anaesthesia, survival

## Abstract

(1) Background: Neonates born with oesophageal atresia and/or tracheooesophageal fistula (OA/TOF) are usually born with a multitude of other congenital anomalies, which may affect their anaesthetic care and survival to hospital discharge. We reviewed the profile and outcome of neonates with OA/TOF and associated congenital cardiac anomalies presenting for surgery at an academic hospital in South Africa. (2) Methods: A retrospective, cross-sectional analysis of all medical records of neonates who had undergone surgical repair of OA/TOF was conducted at an academic hospital between January 2015 and December 2019. Descriptive statistics were used to report the data. Comparisons in perioperative events and outcomes between those with cardiac lesions and those without were done. (3) Results: Neonates presenting for OA/TOF repair with congenital cardiac defects had an incidence of 62 [95% CI 38.5–99.8] per 1000 days since birth. In total, 45.9% had associated cardiac anomalies, with PDA and ASD as the most prevalent lesions. There were statistically significant differences in intraoperative adverse events seen in neonates with congenital cardiac lesions as compared to those without. (4) Conclusion: Despite advances in neonatal critical care and anaesthetic and surgical techniques, OA/TOF associated with congenital cardiac anomalies is still associated with high mortality rates in developing countries.

## 1. Introduction

Oesophageal atresia (OA) with or without tracheoesophageal fistula (TOF) is a rare congenital anomaly characterized by failure of the oesophagus to develop normally [1]. The oesophagus ends blindly as a pouch, either in the neck or the thorax, and an abnormal communication between the trachea and the oesophagus often exists [2]. The incidence of TOF occurs in 1:3000–5000 live births globally [3,4]. There is a 30–50% incidence of associated congenital defects, including vertebral anomalies, anal atresia, congenital heart disease (CHD) and renal and limb anomalies (VACTERL) in neonates with OA and TOF [2]. Cardiac lesions are said to occur in 20% of neonates, while vertebral, anal, renal and limb anomalies occur in 17%, 12%, 16% and 10%, respectively [3].

Surgical repair of OA/TOF is not only considered one of the most difficult procedures, but it requires urgent intervention to prevent aspiration and respiratory compromise. This procedure is usually performed in the first few days of life of a neonate. Anaesthesia for tracheoesophageal fistula repair is one of the challenging conditions that the anaesthesiologist must deal with during the neonatal period. Respiratory distress syndrome of prematurity; poor ventilation due to placement of the tracheal tube above the fistula; pre-existing lung disease secondary to previous aspiration of gastric contents or pathophysiology arising from associated anomalies, especially cardiac lesions, are some of the adversities encountered during anaesthetic management of these neonates [5].

The OA/TOF anomalies are classified according to their anatomic configuration. There are two classifications that are used extensively, namely Gross and Vogt [6]. The most common form of TOF according to Gross is type C and according to Vogt is type IIIb, representing more than 86% of all cases [7]. Type C/IIIb is described as a blind upper oesophageal pouch and distal oesophagus that connects to the trachea via a fistula tract, typically on the posterior aspect near the carina. Type A/II is the second-most prevalent, with 7% of all cases. Type A/II are more complicated, usually requiring correction in stages, taking longer to repair and having higher complication rates compared to type C/IIIb [3,6]. 

In earlier years, TOF repair was characterized by high mortality rates [6]. Several risk factors have been identified over the years, with cardiac lesions dominating in adverse outcomes [2,4,6,8,9,10,11]. 

In Africa, as in the rest of the world, there have been significant advances in neonatal critical care, surgical techniques and neonatal anaesthesia. However, studies examining perioperative management and outcomes of OA/TOF have largely been conducted in developed countries. The current study reviewed the profile and outcome of neonates with tracheoesophageal fistula with associated congenital cardiac anomalies, presenting for surgery at an academic hospital in South Africa.

## 2. Materials and Methods

A retrospective cross-sectional review of neonates booked for repair of a tracheoesophageal fistula was conducted at an academic hospital from 1 January 2015 to 31 December 2019. Approval to conduct the study was obtained from the Wits Human Research Ethics Committee (Medical), the Graduates Studies Committee of the University of the Witwatersrand and the Chief Executive Officer (CEO) of the specific hospital. The study population consisted of all neonates that were booked for tracheoesophageal fistula repair at this hospital in the period between 1 January 2015 and 31 December 2019. 

Data were collated from theatre registers, paper-based medical records from paediatric cardiology and electronic databases from the surgery department. Those cases who were cancelled on preoperative assessment for other medical reasons were excluded. A total of 51 neonates were recorded in the theatre register, and only 37 complete medical records were found. Records with missing significant data were excluded. 

The following data were collected and captured onto an electronic data sheet: demographics; echocardiogram findings; surgical data; anaesthetic data; related preoperative, intraoperative and postoperative critical events and survival to hospital discharge, including in hospital mortality.

## 3. Results

A total of 37 neonates were included in the study (Figure 1). Males and females were evenly distributed, with a ratio of 20:17, respectively (Table 1). Equal proportions of premature and normal births were observed. The majority (64.9%) of both the term and premature neonates weighed less than 2.5 kg and, thus, were considered underweight. Gross Type C was the most common form of TOF (81%) among the study neonates (Figure 2).

The surgical technique commonly employed was a right thoracotomy in 36 neonates, whereas one neonate had bilateral thoracotomies, and none had thoracoscopic repair of OA/TOF.

Amongst the 17 neonates who experienced perioperative complications, 11 had multiple cardiac lesions (Table 2). The mortality rate was 33.3% in the group with single cardiac lesions and 45% in those with multiple lesions.

A total of 22 neonates had surgical repair of the TOF alone, whilst 15 neonates had TOF repair and other related procedures within the index admission in hospital. Some procedures were related to preoperative optimisation, such as gastroscopies and percutaneous endoscopic gastrostomies, whilst others were performed to manage postoperative complications. Three of the neonates had gastrostomy/enterectomy feeding tube placement and ligation of fistula as the initial surgery prior to definitive TOF repair. Six had oesophagoscopy/gastroscopy and dilatation, and two had colostomies, whilst two had Nissen fundoplication and gastrostomies for post-OA/TOF repair complications. One neonate had a colectomy post-OA/TOF repair and demised after that procedure. This neonate had all components of VACTERL. A neonate who had a Hartman’s procedure prior to an OA/TOF repair demised post-repair and was thought to have a syndrome with two vaginas and Prune belly.

One neonate with a cardiac lesion died on the theatre table. Four other neonates with cardiac lesions who demised had nosocomial infection postoperatively. A fifth mortality of a neonate with a cardiac lesion was related to pulmonary hypertension and a 14-day ICU admission. The seventh neonate with a cardiac lesion had unexplained anaemia and hypotension leading to mortality.

Of the two non-cardiac neonates who demised, one succumbed to sepsis and acute kidney injury almost a month postoperatively, whilst the second demised on the surgical table due to delays in identifying and ligating a fistula. 

Neonates with cardiac lesions experienced difficult intubation and hypotension. They had presented with chronic lung disease and pneumonia (*p* = 0.05) (Table 3). Mortality was four-fold in those with cardiac lesions (*p* = 0.034). 

Preterm neonates presented significantly with anaemia (Table 4). Desaturation of >10% from baseline and hypotension occurred predominantly in term neonates.

There were no differences in the pre-operative critical, intraoperative and postoperative critical events and complications between those with single and multiple congenital cardiac abnormalities (Table 5). 

The median length of hospital stay was longer in the group with cardiac abnormalities; however, the difference was not statistically significant. The risk stratification according to the Spitz classification showed a predominance of Group I (65%) followed by Group II (29%) and only one (6%) in Group III. Four neonates classified as Spitz Group I demised, as did three neonates in Spitz Group II (Figure 3). 

There was a univariable association between mortality and having a cardiac lesion, whereas no relationship was found with prematurity in a univariable logistic regression analysis (Table 6).

## 4. Discussion

The majority of the neonates presenting for OA/TOF had cardiac lesions. Other non-cardiac anomalies ranged from 2.6–16.2%. A third of those with a cardiac lesion were premature (<37 weeks). These cardiac defects were predominantly endocardial cushion defects in nature. The frequency of the cardiac conditions in this population is in line with that reported in previous studies and reports [2,12]. The study revealed that at this hospital, thoracotomy is the commonly used surgical approach for OA/TOF repair. It is evident that pre-operative echocardiography is essential to identify any potential cardiac pathology which is associated with high mortality. This is to mitigate unanticipated intra-operative complications.

Waterston et al., in 1962, introduced a prognostic classification, which looked at the following characteristics: weight, pneumonia and associated congenital anomalies against the backdrop of high mortality rates associated with OA/TOF repair in the earlier years of the 21st century [6]. The contribution of these congenital anomalies to the ultimate prognosis is widely accepted. Mortality in these neonates is almost confined to those with complex congenital cardiac anomalies and/or an extremely low birth weight [13,14]. This was similarly evident in our data. Using Spitz classification, the survival rate was worst in the presence of both a major CHD and prematurity. Furthermore, CHD was a greater contributor to this poor outcome when compared to prematurity in this population. However, it is difficult to assume this generalization due to the wide confidence interval, which likely occurred due to our small sample size.

Neonates with congenital cardiac lesions experienced difficulties in intubation, desaturation, chronic lung disease (CLD) and intraoperative hypotension. It is thought that there is a strong relationship between the embryological development of the cardiac and respiratory structures, so that an abnormality in one may be associated with an abnormality in the other [15]. Additionally, the presence of CHD and an associated genetic disorder should alert the clinician to a high likelihood of an airway abnormality [16]. Furthermore, neonates with CHD may be more vulnerable to episodes of desaturation and hypoxia due to the accompanying CHD or pre-existing episodes of aspiration associated with the OA/TOF [17,18]. We also postulate that, due to haemodynamic challenges related to cardiac anomalies, coupled with significant hypoxia experienced, anaesthesiologists may have struggled with intubation, which necessitated accurate placement of the tube in relation to the fistula.

The deformity is usually repaired with the patient in a left lateral decubitus position via a right-sided thoracotomy or thoracoscopic approach [19]. This is associated with a significant ventilation/perfusion mismatch, which may lead to significant right to left shunting, particularly in those with a PDA [20]. There is normally surgical retraction and compression of the lung and heart, worsening the occurrence of the hypoxemia and hypotension.

The majority of the neonates with endocardial cushion defects were likely to experience changes in their shunt direction related to changes in vascular resistance in the pulmonary circulation [21]. This may drive decreases in blood pressure. Anaesthetic induction agents may also reduce systemic vascular resistance and lead to hypotension [22]. Neonates with congenital cardiac lesions in this group experienced respiratory related complications, including difficulties during intubation, which may be exacerbated by prematurity. 

OA/TOF surgery usually needs to be performed urgently to prevent aspiration and respiratory complications. This typically takes place within the first couple of days [23]. The failure to identify and ligate the fistula early leads to gastric distention, hypoventilation, hypoxia, cardiac compromise and death [3,13]. In our cohort, there was delayed presentation for surgery. The cause is often multifactorial. There is also a limitation of resources, such as the availability of postoperative intensive care unit beds. 

The overall mortality in this study was high (25%) compared to that often quoted in contemporary literature, which has substantially decreased to 6–9% in OA/TOF neonates [2,12,24]. This high mortality rate is comparable to periods dating between 1960 and 1979 [11]. In 2022, a multicenter study looking at 298 North American centres by Keefe et al. [25] demonstrated an in-hospital mortality of 12.7%. Another North American study by Wang B et al. [11] confirmed that African American infants associated with low-income status had a lower survival rate compared with white infants, of 84% versus 93% (*p* < 0.001). The study by Vukadin M et al. [26] confirmed that survival in developing countries is much lower and quoted a mortality rate of 28.3%. There is little published data on neonates born with OA/TOF in low-middle income countries where the paucity of financial and human resources strains management and influences outcomes.

Long term morbidities, such as respiratory complications, the recurrence of TOF and anastomotic strictures, are important considerations; however, they were not the focus of this study and will hopefully be addressed in future studies.

In our institution, prenatal scanning for congenital abnormalities is not routine. This may be the reason why our neonates were operated on later, at the median age of 6.5 days. Late diagnosis leads to complications related to aspiration, malnutrition, pneumonia and adverse perioperative outcomes [7]. The late diagnosis and complications related to hypoxia may have led to delayed functional closure of the ductus arteriosus in some of the full-term neonates, which should close within 12 to 24 h [27].

Delays may also be attributed to preoperative patient stabilization and optimization. Some of the neonates often required preoperative ventilation, transfusion and correction of acid–base derangements. There is also a limitation on resources such as the availability of preoperative echocardiography and theatre time.

### Limitations

The study was limited by the sparsity of data available. In the period between 2015 to 2019, 51 neonates born with TOF/OA were identified at a hospital that services an average of 20,000 live births a year. Tracheoesophageal fistula is a rare disease and as a result, the surgical records at this academic hospital showed an average of 10 cases of TOF repair per annum. Furthermore, this study could not compare perioperative critical events between ductal-dependent and non-ductal-dependent groups amongst those with cardiac disease due to data sparsity. The retrospective nature and small sample size of the study limited its data analysis capability. Data management at our hospital is mostly a paper-driven filing system, leading to a reduction of potential study cases due to missing relevant medical records. 

## 5. Conclusions

Despite advances in neonatal critical care and anaesthetic and surgical techniques, oesophageal atresia with or without tracheoesophageal fistula associated with other congenital cardiac anomalies is still associated with high mortality rates in developing countries. It is noted that data on earlier mortality rates in these countries are sparse or unavailable to compare progress made thus far; however the available global data shows better outcomes. The presence of a congenital cardiac anomaly, prematurity and low birth weight in TOF/OA remains strongly related to morbidity and mortality. 

It is imperative for the anaesthetist in their preoperative assessment to have a thorough understanding of underlying congenital heart disease, including anatomy and physiology, and to identify additional risks prior to anaesthetising these neonates, particularly in complex surgery, such as for TOF/OA.

## Figures and Tables

**Figure 1 children-09-00887-f001:**
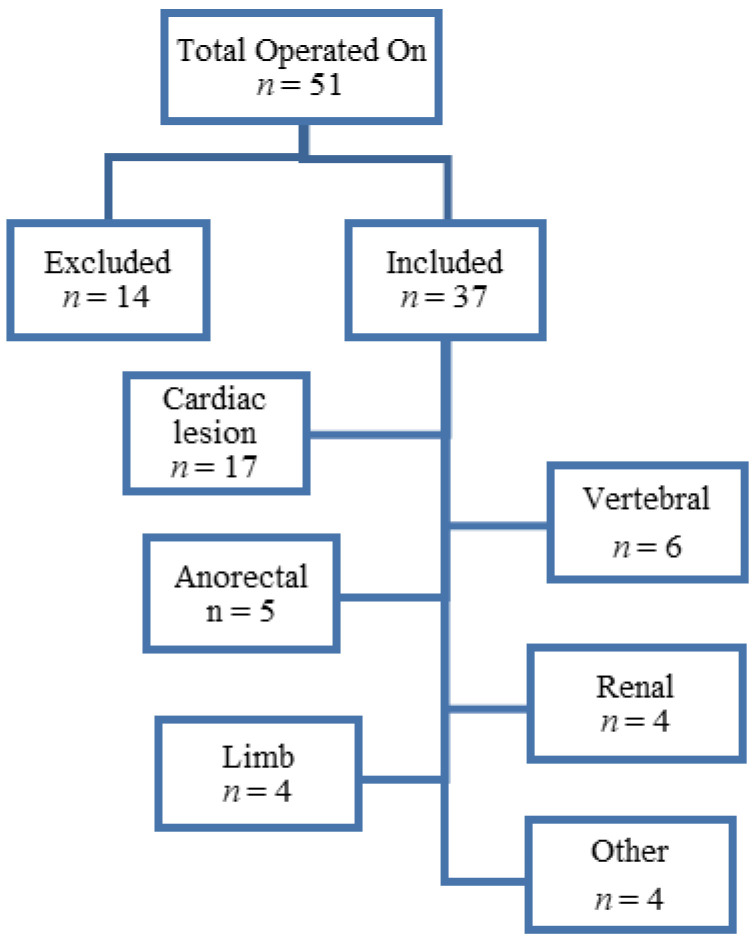
Study flow diagram.

**Figure 2 children-09-00887-f002:**
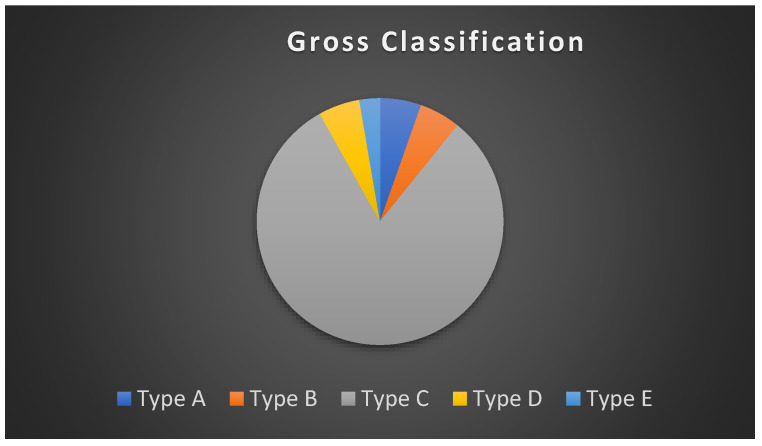
Distribution of tracheoesophageal fistula types among study neonates.

**Figure 3 children-09-00887-f003:**
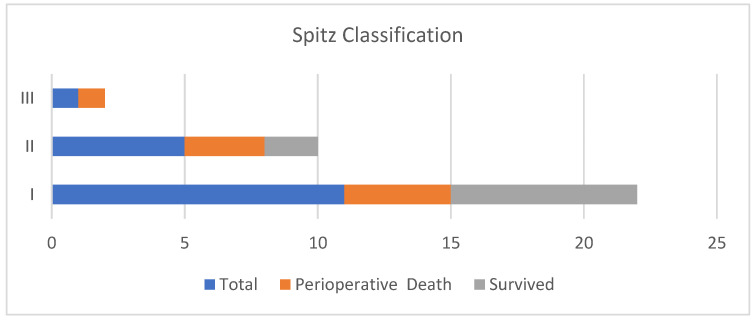
Spitz classification and outcome.

**Table 1 children-09-00887-t001:** Neonates demographics (*n* = 37).

Characteristic		*n* (%)
Sex	Female	17 (46.0%)
	Male	20 (54.0%)
Gestational age (weeks), median (IQR)	Total	36.5 (34.5–40)
	<37	18 (48.7%)
	≥37	18 (48.7%)
	Missing	1 (2.7%)
Age at surgery (days), total median (IQR)		6.5 (3–10.5)
Birth weight (g), mean ± SD	Total	2327 ± 626
	<2500	24 (64.9%)
	≥2500	12 (32.4%)
	Missing	1 (2.7%)
Weight at surgery (g), median (IQR)		2366 (1575–3105)
Proportions of weight groups	<2500	23 (62.2%)
	≥2500	13 (35.1%)
	Missing	1 (2.7%)

IQR—interquartile range, SD—standard deviation.

**Table 2 children-09-00887-t002:** Congenital abnormalities among neonates presenting for TOF repair.

Type of Abnormality		*n* (%)
Vertebral		6 (16.2%)
Anorectal		5 (13.5%)
Cardiac		17 (45.9%)
Renal		4 (10.8%)
Limb		4 (10.8%)
Other	Genitalia	2 (5.3%)
	Choanal atresia	1 (2.6%)
	Pulmonary artery malformation	1 (2.6%)
Echocardiography		*n* (%)
Number of neonates who had echocardiography		30 (81.1%)
Presence of cardiac lesions		17 (45.9%)
Cardiac lesion types	Tetralogy of Fallot	1 (5.2%)
	Atrial septal defect	10 (52.6%)
	Ventricular septal defect	6 (31.6%)
	Patent ductus arteriosus	10 (52.6%)
	Pulmonary hypertension	2 (10.5%)
	Bidirectional shunt	3 (15.8%)
	Tricuspid regurgitation	3 (15.8%)
	Dextroposition	2 (10.5%)

**Table 3 children-09-00887-t003:** Perioperative complications and outcomes between those with and without congenital cardiac abnormalities.

Complications and Outcomes	Overall	No Cardiac Abnormalities	Had Cardiac Abnormalities	*p* Value
Desaturation	13 (35.1%)	6 (30.0%)	7 (41.2%)	0.478
Mechanical ventilation	11 (29.7%)	6 (30.0%)	5 (29.4%)	0.969
Inotropes used or increased	3 (8.1%)	1 (5.0%)	2 (11.8%)	0.452
Invasive monitors	11 (29.7%)	6 (30.0%)	5 (29.4%)	0.545
Difficult intubation	4 (10.8%)	0 (0.0%)	4 (23.5%)	0.022
Difficult ventilation	15 (40.5%)	7 (35.0%)	8 (47.1%)	0.457
Desaturation > 10% from baseline	17 (45.9%)	7 (35.0%)	10 (58.8%)	0.147
Hypotension	9 (24.3%)	2 (10.0%)	7 (41.2%)	0.028
Inotropes used or increased	4 (10.8%)	1 (5.0%)	3 (17.7%)	0.315
Arrythmias	2 (5.4%)	1 (5.0%)	1 (5.9%)	0.906
Blood transfusions used	10 (27.0%)	5 (25.0%)	5 (29.4%)	0.763
Mechanical ventilation (days) Median (IQR)	13.5 (7–30)	14.5 (6–30)	12 (7–30)	0.959
Nosocomial infection	22 (59.5%)	14 (70.0%)	8 (47.1%)	0.157
Strictures	6 (16.2%)	4 (20.0%)	2 (11.8%)	0.498
Septic shock	4 (10.8%)	2 (10.0%)	2 (11.8%)	0.863
Anaemia	5 (13.5%)	3 (15.0%)	2 (11.8%)	0.774
Chronic lung disease	3 (8.1%)	0 (0.0%)	3 (17.7%)	0.050
Pneumonia	4 (10.8%)	4 (20.0%)	0 (0.0%)	0.050
Sepsis	3 (8.1%)	1 (5.0%)	2 (11.8%)	0.452
Length of stay ICU	22.5 (11–36)	16.5 (12–35)	30 (9.5–50)	0.378
Length of stay hospital	37 (18–64)	39.5 (30–60)	36 (14–86)	0.729
Mortality	9 (25.0%)	2 (10.5%)	7 (41.2%)	0.034

ICU—intensive care unit, IQR—interquartile range.

**Table 4 children-09-00887-t004:** Perioperative complications and outcomes between premature and term births.

Variable Median (IQR)/*n*%	Overall (*n* = 36)	Term Birth (≥37 weeks), (*n* = 18)	Prematurity (<37 weeks), (*n* = 18)	*p* Value
Preoperative critical events				
Desaturation	13 (36.1%)	5 (27.8%)	8 (44.4%)	0.489
Mechanical ventilation	11 (30.6%)	3 (16.7%)	8 (44.4%)	0.146
Inotropes used or increased	4 (11.1%)	2 (11.1%)	2 (11.1%)	1.000
Invasive/non-invasive	11 (30.6%)	3 (16.7%)	8 (44.4%)	0.146
Intraoperative critical events				
Difficult intubation	4 (11.1%)	3 (16.7%)	1 (5.6%)	0.603
Difficult ventilation	14 (38.9%)	10 (55.6%)	4 (22.2%)	0.086
Desaturation > 10% from baseline	16 (44.4%)	12 (66.7%)	4 (22.2%)	0.018
Hypotension	8 (22.2%)	7 (38.9%)	1 (5.6%)	0.041
Arrythmias	2 (5.6%)	0 (0.0%)	2 (11.1%)	0.486
Blood transfusions used	10 (27.8%)	5 (27.8%)	5 (27.8%)	1.000
Postoperative critical events and Complications				
Mechanical ventilation (days)	13 (7–30)	12.5 (6–30)	14 (7–25)	0.814
Nosocomial infection	22 (61.1%)	11 (61.1%)	11 (61.1%)	1.000
Strictures	5 (13.9%)	3 (16.7%)	2 (11.1%)	1.000
Septic shock	4 (11.0%)	0 (0.0%)	4 (22.2%)	0.104
Anaemia	5 (13.9%)	0 (0.0%)	5 (27.8%)	0.045
Chronic lung disease	2 (5.6%)	1 (5.6%)	1 (5.6%)	1.000
Pneumonia	4 (11.0%)	2 (11.1%)	2 (11.1%)	1.000
Sepsis	3 (8.3%)	1 (5.6%)	2 (11.1%)	1.000
Mortality	9 (25.7%)	5 (29.4%)	4 (22.2%)	0.711

**Table 5 children-09-00887-t005:** Perioperative complications and outcomes between neonates with single and multiple congenital cardiac abnormalities.

Variable Median (IQR)/*n* (%)	Overall *n* = 17	Single Cardiac Lesion (*n* = 6)	Multiple Cardiac Lesions (*n* = 11)	*p* Value
Preoperative and critical events				
Desaturation	7 (41.2%)	1 (16.7%)	6 (54.6%)	0.304
Mechanical ventilation	5 (29.4%)	1 (16.7%)	4 (36.4%)	0.600
Inotropes used or increased	3 (8.1%)	1 (16.7%)	2 (18.2%)	1.000
Invasive/non-invasive	5 (29.4%)	1 (16.7%)	4 (36.4%)	0.600
Intraoperative and critical events				
Difficult intubation	4 (23.5%)	1 (16.7%)	3 (27.3%)	1.000
Difficult ventilation	8 (47.1%)	2 (33.3%)	6 (54.6%)	0.620
Desaturation > 10% from baseline	10 (58.8%)	2 (33.3%)	8 (72.7%)	0.162
Hypotension	7 (41.2%)	2 (33.3%)	5 (45.5%)	1.000
Arrythmias	1 (5.9%)	1 (16.7%)	0 (0.0%)	0.353
Blood transfusions used	5 (29.4%)	1 (16.7%)	4 (36.4%)	0.600
Postoperative critical events and complications				
Mechanical ventilation (days)	12 (7–30)	25 (12–60)	9 (5–30)	0.099
Nosocomial infection	8 (47.1%)	2 (33.3%)	6 (54.6%)	0.620
Strictures	2 (11.8%)	1 (16.7%)	1 (9.1%)	1.000
Septic shock	2 (11.8%)	1 (16.7%)	1 (9.1%)	1.000
Anaemia	2 (11.8%)	1 (16.7%)	1 (9.1%)	1.000
Chronic lung disease	3 (17.7%)	2 (33.3%)	1 (9.1%)	0.515
Pneumonia	0 (0.0%)	0 (0.0%)	0 (0.0%)	-
Sepsis	2 (11.8%)	0 (0.0%)	2 (18.2%)	0.515
Mortality	7 (41.2%)	2 (33.3%)	5 (45.5%)	0.627

**Table 6 children-09-00887-t006:** Relationship between mortality and cardiac lesions and prematurity.

Parameter	uOR (95%CI)	*p* Value
Cardiac Lesions	6.758 (1.089–41.938)	0.040
Prematurity	1.087 (0.199–5.935)	0.923

CI—confidence interval.

## Data Availability

Due to issues of confidentiality dictated by the institutional ethics committee (HREC), the raw data that were used in this study are not openly available. A request for access would be needed.

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
