# Peer review of "The Phenotypical Profile and Outcomes of Neonates with Congenital Tracheoesophageal Fistula Associated with Congenital Cardiac Anomalies Presenting for Surgery"

_children, 2022, doi:10.3390/children9060887_

Round 1

Reviewer 1 Report

This is an article about the outcome of children with esophageal atresia and congenital heart diseases in a developing country. The overall mortality rate is still high in this study. Congenital heart defects are a further burden for these children.

The study is well written. I just have some minor questions and remarks.

Line 28: As esophageal atresia is a rare disease, you might cancel the word “common”.

Line 97: Could you give more data on the other procedures performed in these 15 children? Did these surgeries have an impact on the outcome?

Could you please explain “difficult intubation” further? Was it a mechanical / anatomical problem or something else? – Do you see any reason why it is found more often in children with congenital heart defects?

Line 105: The differences between those children with cardiac problems and without were mostly trends, but not significant, especially for those related to cardiac problems as desaturation or arrhythmias.

Line 107: which p-value is right that of the text (0.34) or in table 3 (0.034)?

124-126 should be placed within results.

145-148 should be placed within results. Line 145: did these cardiac arrests lead to death in all cases?

Could you differentiate mortality between those children with prematurity (P) and congenital heart defect (CHD) and those with congenital heart defect only? (Like P+, CHD+/ P+ CHD- / P-, CHD+, P-, CHD-)

Author Response

Please see uploaded document.

Reviewer 2 Report

Despite the limitations of the above study as highlighted by the authors, namely limited numbers, high percentage of missing case notes, on balance I recommend acceptance for publication.

I suggest the following amendments:

  1. This should be limited to facts only. The hospital name should not appear here. The subject material should be OA+/- TOF and not the other way round. References should not be cited.
  2. OA and TOF are not common anomalies (incidence of one in 3/5,000 live births).
  3. 51 cases admitted over a 5 year period of which 17 case notes were missing leaves a total of 37 cases to be analysed. There were 17 cardiac anomalies, 6 being single anomalies with 2 deaths and 11 with multiple anomalies with 5 deaths. This is not statistically different. The difficulty with intubation, ventilation, and cardiac arrest in the perinatal period is an anaesthetic problem!

Reviewer 3 Report

This study aims to assess the impact of congenital cardiac anomalies on the outcome of oesophageal atresia.

In general, the paper is clearly and well written. The main limitation of the study, in my opinion, is small sample size. Regardless, the results are clear, although somewhat expected.

Comments:

At least according to EUROCAT classification (https://eu-rd-platform.jrc.ec.europa.eu/sites/default/files/Section%203.2-%2027_Oct2016.pdf) PDA should not be considered as a congenital anomaly in premature neonates (<37 weeks of gestation). Was this taken into account?

Did the authors consider using Spitz classification for assessing the risk for mortality?

The median age at operation was 6.5 days. What was the cause of delay as usually OA/TOF patients are operated within the first 24 h of life?

How many patients were born at the operating hospital and how many were transferred?

In my opinion, the perioperative complications should be classified using Clavien-Dindo grading.

Number of VACTERL patients? Were their outcomes inferior compared to non-VACTERL cases?

Minor comments:

In general, term “OA/TOF” is used more commonly than "TOF/OA", but that is just a matter of presentation. If the authors prefer TOF/OA, I do not oppose that.

The authors state in Introduction (line 28) that TOF is a common anomaly and then later in Limitations (line 157) state that TOF is rare. 

Author Response

Please see uploaded document.

Round 2

Reviewer 3 Report

The authors have answered well to the majority of my concerns. However, I still feel that adding Clavien-Dindo grading to complications would improve the quality of the manuscript.

Author Response

Please see uploaded document.
